# Predictive Value of Prognostic Nutritional Index for Early Postoperative Mobility in Elderly Patients with Pertrochanteric Fracture Treated with Intramedullary Nail Osteosynthesis

**DOI:** 10.3390/jcm12051792

**Published:** 2023-02-23

**Authors:** Leon Marcel Faust, Maximilian Lerchenberger, Johannes Gleich, Christoph Linhart, Alexander Martin Keppler, Ralf Schmidmaier, Wolfgang Böcker, Carl Neuerburg, Yunjie Zhang

**Affiliations:** 1Department of Orthopaedics and Trauma Surgery, Musculoskeletal University Center Munich (MUM), University Hospital, LMU Munich, 81377 Munich, Germany; 2Department of Medicine IV, Geriatrics, University Hospital, LMU Munich, 80336 Munich, Germany

**Keywords:** orthogeriatrics, pertrochanteric fracture, nutrition, prognostic nutritional index, TFNA, mobility

## Abstract

Background: Early postoperative mobilization is essential for orthogeriatric patients. The prognostic nutritional index (PNI) is widely used to evaluate nutritional status. This study sought to investigate the predictive value of PNI for early postoperative mobility in patients with pertrochanteric femur fractures. Materials and methods: This study included 156 geriatric patients with pertrochanteric femur fractures treated with TFN-Advance™ (DePuy Synthes, Raynham, MA, USA). Mobility was evaluated on the third postoperative day and by discharge. Stepwise logistic regression analyses were performed to evaluate the association significance of PNI with postoperative mobility together with comorbidities. The optimal PNI cut-off value for mobility was analyzed using the receiver operating characteristic (ROC) curve. Results: Three days postoperatively, PNI was an independent predictor of mobility (OR: 1.14, 95% CI: 1.07–1.23, *p* < 0.01). By discharge, it was found that PNI (OR: 1.18, 95% CI: 1.08–1.30, *p* < 0.01) and dementia (OR: 0.17, 95% CI: 0.07–0.40, *p* < 0.001) were significant predictors. PNI correlated weakly with age (r = −0.27, *p* < 0.001). The PNI cut-off value for mobility on the third postoperative day was 38.1 (specificity = 78.5%, sensitivity = 63.6%). Conclusions: Our findings indicate that PNI is an independent predictor of early postoperative mobility in geriatric patients with pertrochanteric femur fractures treated with TFNA™.

## 1. Introduction

Trochanteric fractures were the second most common fracture among German adults from 2009–2019 after femoral neck fractures, and 87% of patients diagnosed with trochanteric fractures were aged ≥70 [1]. A hazard ratio for one-year mortality of 2.78 was shown in geriatric patients suffering from a hip fracture in comparison to the same age group without a prevalence of hip fracture [2] with a one-year mortality of 30% [3].

Postoperative care should focus on preventing complications and promoting quick mobilization with full weight-bearing as tolerated. Ottesen et al. found that restricted postoperative weight-bearing in geriatric patients with a hip fracture was associated with significantly higher rates of adverse events, such as sepsis, pneumonia, delirium, transfusion, and increased length of hospitalization [4]. A previous study reported that prolonged immobility after hip fracture was related to higher six-month mortality and lower functional levels two months after the event [5].

Using patient-specific factors is a crucial step to identify patients at risk for immobility. An early assessment can lead to therapeutic changes according to the patient’s needs and thereby reduce postoperative morbidity and mortality [6]. The correlation between nutritional status and postoperative outcomes has drawn more attention in the current literature. Ihle et al. found that malnourished geriatric trauma patients showed delayed postoperative mobilization compared to patients with a regular nutritional status [7]. Moreover, they reported an increased prevalence of malnutrition in older trauma patients, as malnutrition was prevalent in roughly 12% of patients aged <65, 31% of patients aged 65–80, and 60% in patients aged >80 [7].

Malnutrition can be evaluated by various methods, for example, with the mini nutritional assessment (MNA) [8], nutritional risk screening (NRS) [9], body mass index (BMI), or laboratory parameters. The prognostic nutritional index (PNI) is a laboratory index based on serum albumin and total lymphocyte count [10]. PNI was initially developed to preoperatively assess perioperative risks in gastrointestinal surgery. In recent research, PNI was shown to be a promising prognostic factor and predictor of postoperative outcomes in different tumor entities such as pancreatic cancer, colorectal cancer, or lung cancer [11,12,13]. Low PNI was found to be a predictive factor for postoperative delirium, infectious complications, and ICU admission in hip fracture patients [14,15]. Geriatric hip fracture patients with hypoalbuminemia were shown to have significantly higher postoperative adverse events and mortality rates compared to patients with normal serum albumin concentration [16,17]. However, the value of PNI in predicting the postoperative mobility of hip fracture patients remains unclear.

This study aims to investigate the prognostic value of the PNI on postoperative mobility in trochanteric hip fracture patients. We hypothesized that patients with low PNI have reduced postoperative mobility.

## 2. Patients and Methods

### 2.1. Patient Selection

The study protocol was approved by the local ethics committee (approval number: 20-0247). Geriatric patients (age ≥ 65 years) suffering from pertrochanteric femoral fractures (ICD-10 code: S72.1, AO: 31A1.2, 31A1.3, 31A2.2, 31A2.3 [18], Evans: Type I [19]) and treated with the TFN-ADVANCED™ Proximal Femoral Nailing System (TFNA, DePuy Synthes, Raynham, MA, USA) consecutively from 1 June 2020 to 1 May 2022 in our university teaching hospital were retrospectively enrolled. Notably, isolated single trochanteric fracture and intertrochanteric (reverse obliquity) fracture were excluded due to the different treatment strategies (AO: 31A1.1, 31A3, and Evans Type II). Five deceased patients were excluded due to their lack of mobility status. Notably, there were no patients with chronic liver dysfunction or end-stage liver disease included in our study.

The surgical procedure can be described briefly as follows: After either general or regional anesthesia, the patients were placed supine on a table with a leg holder for closed reduction. An incision (about 3 cm) was made proximal to the greater trochanter after a successful closed reduction. This was followed by the insertion of the TFNA-Nail after the measurement of intramedullary width. The femoral blade and an anti-rotational screw could be then inserted via 1 cm incisions guided by the provider instruments. After a satisfying intraoperative X-ray control, the wounds were closed and a whole-leg spica bandage was applied.

### 2.2. Data Selection

The historical patient data were retrieved from the inpatient database of our hospital (Meona Ltd., Freiburg, Germany) and irreversibly anonymized before analysis in a confidential database (Microsoft Excel 2018, Microsoft Corporation, WA, USA). Demographic data, including age, gender, and body mass index (BMI), were collected. Preoperative comorbidities such as urinary tract infection (UTI), atrial fibrillation, chronic kidney disease (CKD), dementia, stroke, as well as anesthesia types, status of the American Society of Anesthesiologists (ASA), and operation length, were collected. Postoperative events within 3 postoperative days such as moderate or severe electrolyte disorder (defined as Na^+^ < 135 mmol/L/ > 145 mmol/L and K^+^ < 3.5 mmol/L/ > 5 mmol/L), pneumonia, postoperative anemia requiring blood transfusion, and treatment necessity from the intermediate care (IMC) or the intensive care unit (ICU) were also included.

On the first postoperative day, blood testing was routinely performed in our laboratory institute for postoperative control, including vitamin D levels for osteoporosis diagnosis. The PNI was calculated from these laboratory results as well, using the formula: 10 × albumin value + 0.005 × total lymphocyte count from peripheral blood [10]. All patients received a high-caloric supplement (Fresubin, Bad Homburg, Germany) to compensate for the increased metabolism caused by trauma and operation. Vitamin D deficiency was orally supplemented. Specific osteoporosis therapy such as anti-resorptive therapy or bisphosphonate was not routine during the acute management.

After the surgery, all patients received physiotherapy on the first postoperative day to regain mobility. Pain-adapted full weight-bearing was allowed immediately after surgery for all patients. In case full weight-bearing was not possible, a stepwise mobilization protocol with passive training, repositioning in bed, and assisted mobilization out of bed was performed. The postoperative mobilization achievements were documented daily. Patients who were mobile with or without help such as walking with a forearm walking frame, a rollator, or crutches were defined as mobilizable. Patients documented as lying, sitting, and standing were defined as immobile. The mobility status of patients on the third postoperative day, as well as by discharge was used as the outcome of the current study.

### 2.3. Statistics

The statistical analysis was performed using SPSS version 29 (SPSS Inc., Chicago, IL, USA) and R version 4.0.5. Categorical data were compared using Fischer’s exact test or the Pearson chi-square test and presented as percentages. The Kolmogorov–Smirnov test was performed to verify the normality of quantitative data, which were presented with an average ± standard deviation. If confirmed, the student *t*-test was used to determine the significance; if not confirmed, the Mann–Whitney U test was applied. The Pearson correlation coefficient (r) was used to identify the strength of the correlation. Analysis of variance on ranks followed by the Student–Newman–Keuls method was used to estimate stochastic probability in intergroup comparison.

The receiver operating characteristic (ROC) curve was performed to calculate the optimal cut-off value of PNI for the mobility on the third postoperative day by the highest Youden index. Stepwise regression was used to investigate the risk factors. Univariate logistic regression analyses were performed to filter the relevant independent variables with a *p*-value < 0.1 to be used in the final model. Multivariate logistic regression analyses were then performed for the final evaluation. An odds ratio (OR) greater than 1.0 indicated a higher chance of mobility, whereas an OR less than 1.0 indicated a higher chance of immobility. The area under the curve (AUC) was calculated to examine the performance of the regression models. A two-tailed *p* < 0.05 was considered significant. The post hoc analysis was performed using G-Power [20] (Heinrich-Heine-University, Düsseldorf, Germany).

## 3. Results

A total of 156 patients who suffered trochanteric fractures and underwent surgery using TFNA™ were consecutively enrolled in the current study. The average length of hospital stay was 13.5 ± 6.4 days (ranging from 5–30 days). The probability of 1-ß error was 0.96 using the post hoc analysis. The best cut-off value of PNI to predict patients’ early mobility on the third postoperative day was 38.1 (sensitivity: 63.6%, specificity: 78.5%, and AUC: 0.73) according to the maximum Youden index using ROC.

Using the best cut-off value as a reference, the patients were divided into two groups (Table 1). The result suggested that low PNI was significantly associated with age (*p* = 0.01), postoperative anemia (*p* = 0.01), the necessity of treatment in IMC (*p* = 0.01) or ICU (*p* < 0.01), vitamin D deficiency (*p* = 0.01), UTI (*p* = 0.04), atrial fibrillation (*p* = 0.02), dementia (*p* = 0.02), and operation length (*p* = 0.02). A negative and weak correlation was found between PNI and age (r = −0.27, y = −0.17x + 51.12, *p* < 0.001, Figure 1). PNI and BMI showed no significant correlation (*p* = 0.205).

Univariate regression was first performed to determine the relevant independent prognostic factors for patients’ mobility three days after TFNA™ surgeries (Table 2), from which the following factors were selected for the multivariate logistic regression: PNI (*p* < 0.0001), the necessity of treatment in IMC (*p* = 0.07), or ICU (*p* = 0.03), vitamin D deficiency (*p* = 0.09), UTI (*p* = 0.09), atrial fibrillation (*p* = 0.06), dementia (*p* = 0.02), and stroke (*p* = 0.06). The multivariate logistic regression showed that only PNI (OR: 1.14, 95% CI: 1.07–1.23, *p* < 0.01) was significantly associated with patients’ mobility three days after TFNA™ surgeries. The AUC of this model was 0.80.

Stepwise regression was also performed to evaluate prognostic factors for the final mobility by the end of the stationary therapy (Table 3). PNI (*p* < 0.0001), transfusion (*p* < 0.001), IMC (*p* = 0.08), or ICU (*p* < 0.001) treatment, UTI (*p* = 0.04), and dementia (*p* < 0.0001) were recognized as relevant variables. The multivariate logistic regression showed that PNI (OR: 1.18, 95% CI: 1.08–1.30, *p* < 0.01) and dementia (OR: 0.17, 95% CI: 0.07–0.40, *p* < 0.001) were significantly related to the final mobility by discharge. With each unit increase of the PNI, there was an 18% higher chance for patients to reach mobility, whereas the presence of dementia was associated with an 83% chance of immobility by discharge. The AUC of this model was 0.86.

The means of PNI from patients with different mobility three days after TFNA™ surgeries and by discharge were calculated, and inter-group comparisons were performed (Figure 2). The patients who were able to walk with crutches and forearm walking frames three days after TFNA™ surgeries, as well as by discharge, exhibited significantly higher PNI than immobilized patients. By discharge, the bedridden patients had a significantly lower PNI than patients walking with crutches, rollators, and walking frames.

The means of PNI in patients with different AO classifications were analyzed. No significant inter-group differences were found in different severities of pertrochanteric fractures (Figure 3).

## 4. Discussion

The objective of the present study was to investigate the prognostic value of the PNI on postoperative mobility in patients with trochanteric hip fractures after TFNA™ surgery. Mobility status was evaluated on the third postoperative day and by discharge in our study. In a prospective study analyzing factors influencing early postoperative mobilization, Said et al. found that only 43% of patients with a hip fracture were able to mobilize within 48 h after surgery [21]. The second mobility evaluation was performed by discharge. There is a high level of clinical interest in the patient’s final status at the end of primary inpatient treatment. This might be the milestone for further therapy and the potential necessity of rehabilitation or ambulatory care.

Our main finding indicated that PNI was an independent prognostic factor for mobility three days postoperatively and by discharge. An increment of each unit in PNI was associated with a 14% (OR = 1.14, *p* < 0.001) higher probability for patients to reach mobility on the third postoperative day, and 18% by discharge (OR = 1.18, *p* > 0.001). To our knowledge, this is the first study investigating the prognostic value of PNI to predict postoperative mobility in patients with trochanteric fractures.

Dementia was a significant factor by discharge, as the presence of dementia was associated with an 83% risk of immobility (OR = 0.17, *p* < 0.001). Hou et al. reported concurring results in a systematic review of the effects of dementia on patients undergoing hip fracture surgery [22]. Another study found that dementia was a significant factor for the unsuccessful recovery of pre-fracture walking ability by discharge in geriatric patients with hip fractures [23]. This might be due to the lack of motivation and compliance in demented patients, so they benefited less from the physiotherapeutic training. Interestingly, dementia was not a significant factor three days postoperatively. This finding implicated the fact that the acute re-mobilization shortly after the surgery depended more on the general condition than the cognitive status of the patients. However, cognitive status showed its importance in progress and in the eventual achievement of mobility at the end of the acute medical treatment. Notably, postoperative delirium (POD) was found as one of the significant factors for delayed mobilization [21]. However, POD could develop any time after the surgery, which was thus not included in our analysis as a predictive factor. All demented patients would surely have a consistent POD. Dementia from anamnesis was thus a good alternative with a good predictive perspective.

Patients with a PNI below the cut-off had a significantly higher prevalence of ambulant UTI, postoperative anemia, and vitamin D deficiency. Further, low PNI was associated with the necessity of IMC or ICU treatment. This is in line with previous literature, reporting higher rates of postoperative complications, as well as a higher incidence of UTI and vitamin D deficiency in patients with low PNI [12,13,14,15]. Interestingly, our result showed that a low PNI was not associated with a higher complexity of the pertrochanteric fracture. Similar findings were described in that there was no association between the nutrition status and the severity of hip or radius fracture [24,25]. Currently, evidence on the prognostic role of PNI regarding intraoperative complications is lacking. No significant correlation could be found between BMI and PNI in the current study. This was an interesting finding suggesting that the BMI alone was misleading to reflect the real nutritional status. This was also in line with the recent opinion that obesity and malnutrition could coexist. The fat accumulation could cause nutritional derangement, affecting the nutritional status negatively both directly through metabolic change and indirectly through chronic or acute diseases [26,27].

Mean PNI was significantly lower in patients standing, sitting, or lying on the third postoperative day than in patients mobilized with crutches or a rollator. A trend was found in the current study: The higher the PNI value the better the postoperative mobilization might be. Patients with higher PNI tended to mobilize themselves more independently. By the time point of discharge, great numbers of patients made progress and were relocated to the better mobility groups. The averages of PNI were still higher in these groups compared to the groups with immobility. This implied that a greater expectation of independent mobility could be given to patients with higher PNI before discharging them from primary care even if they could not mobilize themselves well at the very beginning. The final mobility would be an interesting outcome measure. Commonly performed score systems such as fracture mobility score and Parker mobility score are often used to evaluate the 6-month functional outcome and 1-year mortality of patients, especially those with hip fractures [28]. However, these score systems were based on the mobility level after the discharge. Consequently, they cannot serve the acute assessment immediately after the surgery.

Although nutritional status evaluation with PNI is widespread in the surgical region for over 40 years [10,29,30], malnutrition can be assessed by various alternative methods. There is no defined gold standard for malnutrition assessment. The geriatric nutritional risk index (GNRI), based on albumin and BMI, and the controlling nutritional status score (CONUT), based on lymphocyte count and albumin, are two alternative tools [31,32]. In comparison, PNI is easier for the physician to collect, with only simple scores from routine laboratory tests. Other methods such as MNA and NRS are screening tools based on patient data (BMI, gender, age) and questionnaires covering weight loss, eating habits, and medical history [8,9]. The reliability of answering questionnaires as a part of MNA and NRS deviates from demented patients, who could be commonly found in the geriatric patient group.

A few limitations of this study should be recognized. First, it is a single-center study, data was raised retrospectively. The sample size is rather small. However, the recruited patients were all geriatric patients with trochanteric fractures and identically received TFNA™. Moreover, PNI was calculated based on postoperative laboratory results, while practically, preoperative laboratory results might serve as a better patient screening method [10]. Due to the retrospective study design, the parameters required for PNI calculation could only be collected postoperatively. Surgery may alter postoperative lymphocyte count or albumin levels. A decrease in lymphocytes in the peripheral blood after surgery due to redistribution toward lymphatic tissue has been described by Toft et al. [33]. Therefore, postoperative PNI calculation in the current study may lead to a systemic lowering of the values.

Our findings suggest that malnutrition, assessed by PNI, could lead to reduced postoperative mobility after trochanteric fracture treatment with TFNA™. This is consistent with previous research [7,17]. It raises the question, of whether perioperative nutritional supplementation might have a positive effect on postoperative mobility in geriatric patients. Recent literature offers inconsistent results on the benefit of nutritional supplementation. A retrospective study examined the effect of oral nutritional supplementation (ONS) with enriched formula after hip surgery and reported no significant reduction of postoperative complications and mortality [34]. Williams et al. found a significantly reduced length of hospital stay (LOS) in elderly patients, who received early postoperative ONS, after hip fracture treatment [35]. However, a recent systematic review of five randomized controlled trials on the effect of preoperative ONS in hip fracture patients found a significantly lower risk of postoperative complications but no significant difference in LOS [36]. The malnourished status in geriatric patients might be a consequence of multiple factors. For example, mal-resorption due to chronic gastritis [37], swallowing disorder [38], advanced liver diseases [39], or the diet itself could all contribute to chronic malnutrition. Simply improving oral intake might not be sufficient for this complex situation. More interdisciplinary effort should be given to ensure an adequate nutritional status of the geriatric patient before the injuries ever happen, which was also proved to be a good preventive method [40].

In short, the present data suggested that PNI was an independent, significant factor to predict postoperative mobility in patients treated with TFNA™ after trochanteric femur fracture.

## Figures and Tables

**Figure 1 jcm-12-01792-f001:**
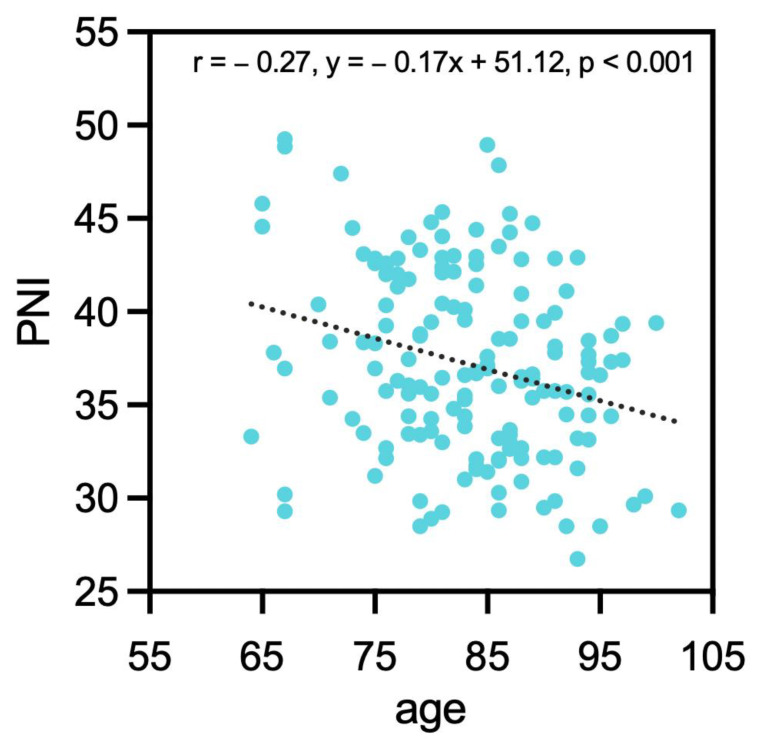
Correlation between PNI and age. PNI: prognostic nutrition index.

**Figure 2 jcm-12-01792-f002:**
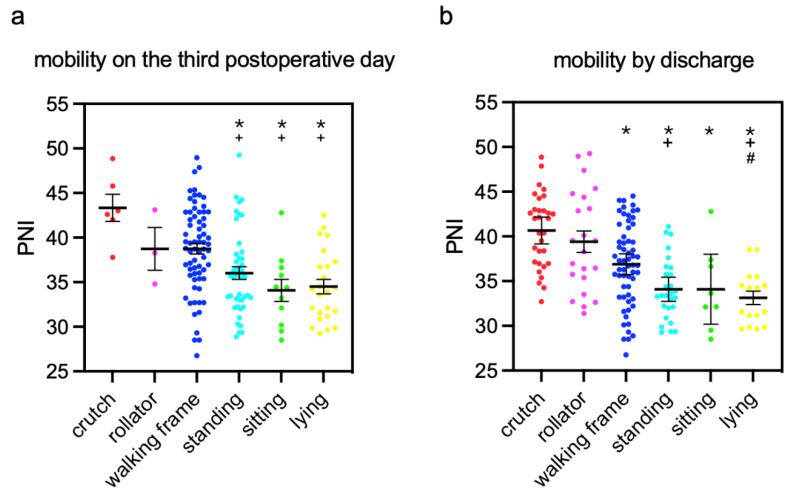
Means of PNI in patients with different extents of mobility (**a**) three days after the surgery and (**b**) by discharge. mean ± standard error of the mean *: *p* < 0.05, compared to PNI in patients who could walk with crutches, +: *p* < 0.05, compared to PNI in patients who could walk with a rollator; #: *p* < 0.05, compared to PNI in patients who could walk with a walking frame.

**Figure 3 jcm-12-01792-f003:**
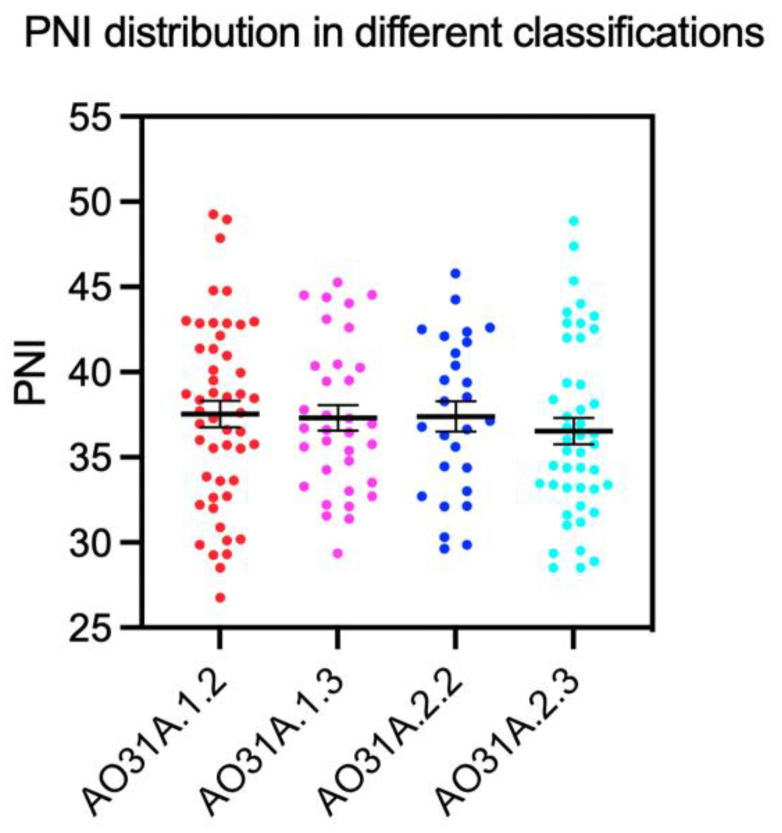
Means of PNI in patients with different extents of fracture classification, mean ± standard error of the mean.

**Table 1 jcm-12-01792-t001:** Clinical factors in patients with low and high prognostic nutritional index.

	Overall (*n* = 156)	Prognostic Nutritional Index		*p*-Value
Factors		<38.1 (*n* = 92)	≥38.1 (*n* = 64)	
Sex (female)	121 (77.1%)	68 (73.9%)	53 (82.8%)	0.22
Age (years)	83.4 (65–102)	84.8 (65–102)	81.5 (65–100)	0.01 *
BMI (kg/m^2^)	23.5 ± 4.4	23.1 ± 4.6	23.9 ± 4.1	0.20
BMI (>24.9 kg/m^2^)	42 (26.9%)	23 (25.0%)	19 (29.7%)	0.52
BMI (18.5–24.9 kg/m^2^)	94 (60.2%)	56 (60.9%)	38 (59.4%)	0.85
BMI (<18.5 kg/m^2^)	20 (12.9%)	13 (14.1%)	7 (10.9%)	0.56
ASA 1	1 (0.64%)	1 (1.1%)	-	1.00
ASA 2	32 (20.5%)	14 (15.2%)	18 (28.1%)	0.05
ASA 3	116 (74.4%)	71 (77.2%)	45 (70.3%)	0.36
ASA 4	7 (4.5%)	6 (6.5%)	1 (1.6%)	0.24
Anesthesia (regional)	53 (33.9%)	27 (29.4%)	26 (40.6%)	0.14
Postoperative anemia	54 (34.6%)	39 (42.4%)	15 (23.4%)	0.01 *
Ward (regular)	62 (39.7%)	22 (23.9%)	40 (62.5%)	0.00 **
Ward (IMC)	72 (46.2%)	50 (54.4%)	22 (34.4%)	0.01 *
Ward (ICU)	22 (14.1%)	20 (21.7%)	2 (3.13%)	0.00 **
Vitamin D deficiency	107 (68.6%)	71 (77.2%)	36 (56.6%)	0.01 *
Electrolyte disorder	45 (28.9%)	28 (30.4%)	17 (26.6%)	0.60
Pneumonia	5 (3.2%)	3 (3.3%)	2 (3.1%)	1.00
UTI	59 (37.8%)	41 (44.6%)	18 (28.1%)	0.04 *
Atrial fibrillation	34 (21.8%)	26 (28.3%)	8 (12.5%)	0.02 *
CKD	40 (25.6%)	26 (28.3%)	14 (21.9%)	0.37
Dementia	28 (17.9%)	22 (23.9%)	6 (9.4%)	0.02 *
Stroke	13 (8.3%)	10 (10.9%)	3 (4.7%)	0.17
Operation length (min)	69.5 ± 41.8	75.9 ± 46.8	60.3 ± 31.1	0.02 *

BMI: body mass index; ASA: status of the American Society of Anesthesiologists; IMC: intermediate care; ICU: intensive care unit; UTI: urinary tract infection; CKD: chronic kidney disease. * *p* < 0.05, ** *p* < 0.01. Percentages in brackets = number of patients with positive factor/total number of patients in the respective column.

**Table 2 jcm-12-01792-t002:** Risk factors for prognosis of mobility on the third postoperative day using stepwise logistic regression.

	Univariate Logistic Regression			Multivariate Logistic Regression		
Factors	OR	95% CI	*p*-Value	OR	95% CI	*p*-Value
Sex (female)	0.90	0.42–1.91	0.78			
Age	0.97	0.93–1.01	0.12			
BMI (>24.9 kg/m^2^)	1.00					
BMI (18.5–24.9 kg/m^2^)	1.01	0.50–2.03	0.98			
BMI (<18.5 kg/m^2^)	1.44	0.52–4.03	0.48			
ASA 1	1.00					
ASA 2	1.13	0.33–1.93	0.80			
ASA 3	0.97	0.50–1.48	0.78			
ASA 4	0.87	0.43–1.35	0.75			
Anesthesia (regional)	1.55	0.80–3.03	0.19			
PNI	1.19	1.10–1.28	0.00 ***	1.14	1.07–1.23	0.00 **
Postoperative anemia	0.74	0.38–1.43	0.37			
Ward (regular)	1.00					
Ward (IMC)	0.37	0.12–1.10	0.07	0.52	0.18–1.38	0.29
Ward (ICU)	0.16	0.05–0.50	0.03 *	0.49	0.16–1.43	0.29
Vitamin D deficiency	0.56	0.28–1.11	0.09	0.62	0.32–1.20	0.23
Electrolyte disorder	0.59	0.29–1.19	0.14			
Pneumonia	0.68	0.11–4.16	0.67			
UTI	0.57	0.30–1.10	0.09	0.89	0.47–1.69	0.77
Atrial fibrillation	0.48	0.22–1.05	0.06	0.65	0.29–1.39	0.35
CKD	0.69	0.33–1.42	0.31			
Dementia	0.28	0.11–0.70	0.03 *	0.41	0.17–0.96	0.09
Stroke	0.28	0.07–1.06	0.06	0.41	0.11–1.35	0.24
Operation length	1.00	1.00–1.01	0.19			

BMI: body mass index; ASA: status of the American Society of Anesthesiologists; PNI: prognostic nutrition index; IMC: intermediate care; ICU: intensive care unit; UTI: urinary tract infection; CKD: chronic kidney disease; OR: odds ratio; CI: confident interval. * *p* < 0.05, ** *p* < 0.01, *** *p* < 0.001.

**Table 3 jcm-12-01792-t003:** Risk factors for prognosis of mobility by discharge using stepwise logistic regression.

	Univariate Logistic Regression			Multivariate Logistic Regression		
Factors	OR	95% CI	*p*-Value	OR	95% CI	*p*-Value
Sex (female)	0.89	0.38–2.08	0.78			
Age	0.93	0.88–0.97	0.00 **	0.98	0.93–1.03	0.07
BMI (>24.9 kg/m^2^)	1.00					
BMI (18.5–24.9 kg/m^2^)	1.39	0.65–2.96	0.39			
BMI (<18.5 kg/m^2^)	3.09	0.80–11.98	0.11			
ASA 1	1.00					
ASA 2	1.11	0.45–3.23	0.87			
ASA 3	0.85	0.24–2.68	0.83			
ASA 4	0.96	0.36–4.35	0.79			
Anesthesia (regional)	0.72	0.35–1.48	0.37			
PNI	1.25	1.14–1.38	0.00 ***	1.18	1.08–1.30	0.00 **
Postoperative anemia	0.33	0.16–0.68	0.00 **	0.54	0.26–1.14	0.17
Ward (regular)	1.00					
Ward (IMC)	0.42	0.10–0.96	0.08	0.45	0.17–1.18	0.17
Ward (ICU)	0.11	0.03–0.34	0.00 ***	0.32	0.10–0.99	0.10
Vitamin D deficiency	0.80	0.37–1.72	0.56			
Electrolyte disorder	0.67	0.32–1.43	0.31			
Pneumonia	0.67	0.32–1.43	0.31			
UTI	0.46	0.23–0.95	0.04	0.80	0.38–1.73	0.63
Atrial fibrillation	0.89	0.38–2.06	0.79			
CKD	0.72	0.33–1.58	0.42			
Dementia	0.11	0.04–0.27	0.00 ***	0.17	0.07–0.40	0.00 ***
Stroke	0.41	0.13–1.29	0.13			
Operation length	1.00	0.99–1.00	0.66			

BMI: body mass index; ASA: status of the American Society of Anesthesiologists; PNI: prognostic nutrition index; IMC: intermediate care; ICU: intensive care unit; UTI: urinary tract infection; CKD: chronic kidney disease; OR: odds ratio; CI: confident interval. ** *p* < 0.01, *** *p* < 0.001.

## Data Availability

The data presented in this study are available on request from the corresponding author.

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
