# Peer review of "Predictive Value of Prognostic Nutritional Index for Early Postoperative Mobility in Elderly Patients with Pertrochanteric Fracture Treated with Intramedullary Nail Osteosynthesis"

_jcm, 2023, doi:10.3390/jcm12051792_

Round 1
Reviewer 1 Report
The acquisition of gait in the increasing number of patients with pertrochanteric fractures is an important topic for orthopedics, and I appreciated and read with interest the analysis of 156 patients' data related to nutrition and postoperative mobility. The result that PNI, which can be tested by blood, was a predictor of early gait acquisition was very interesting. I thought this information is very useful.
You researched patients with AO:31A1-31A3. As I am an orthopedic surgeon and would like to see a breakdown of fracture type in the results. I am interested in the relationship between fracture type and nutrition.
In line 218 of the Discussion, you state that low PNI would result in less bone quality and more complex fractures. If you have the data, I think you should show it. If you can't show the data, I think you should put it in the Limitation.
Line 250, GNRI is calculated from albumin and BMI and does not include lymphocytes.
As you mentioned in the Limitation, you have calculated PNI from postoperative blood. I assume you also checked blood before the surgery, but did you not have that data? Or do you not get the expected results? I would like to know why you used the blood from the first post-op day and I think it should be stated.
Author Response
We thank for all the useful comments and the detailized reply is in the pdf

Reviewer 2 Report
please see the attachment.

Author Response
We thank for all the comments and the meticulous work from the reviewer. We hope our changes can meet the requirement for the publication. For responses please see the pdf file

Round 2
Reviewer 2 Report
Thank you for the revision. The article has obviously been improved, but there are still some problems.
1. How the surgery is done must be described in detail.
2. Sample size analysis or power analysis should be completed, and G power software is recommended for post hock analysis.
3. Does all the data conform to normal distribution?
4. Postoperative mental state will affect postoperative activities of patients, such as delirium in the elderly, which may have adverse effects on postoperative recovery. It is not discussed by the authors.
5. It is not reasonable to calculate PNI by blood test on the first day after surgery. I can understand the condition of the authors that they cannot get the preoperative data. Blood loss during surgery and traumatic stress will lead to changes in blood test indexes of patients; and poor liver function before surgery will also lead to changes in the value of albumin. Ignoring these questions may have resulted in a bias in the results.
6. As written in line 233-234, the OR value of dementia is less than 1 (0.17), indicating that the risk of postoperative immobilization in patients with dementia was lower than patients without dementia, which is a protective factor. Maybe I am not clear about the result categorical variable when the author performed logistic regression.
Author Response
Please find the revision in detail in the pdf. file
thank you

Round 3
Reviewer 2 Report
|
As I mentioned in previous comments, this study has some flaws. But, the authors addressed my comments well. In addition, the authors' research does have some implications, and the limitations of this study may provide directions for future research. I assume that the authors are going to deal with the prognostic nutritional index (PNI) calculated by preoperative data and the results of long-term follow-up after surgery. The English writing must be checked. Thank you again for allow me to review this manuscript. |